# Assessment of Shoreline Changes for the Selangor Coast, Malaysia, Using the Digital Shoreline Analysis System Technique

**Khairul Nizam Abdul Maulud** [1,2,*] **, Siti Norsakinah Selamat** [1] **, Fazly Amri Mohd** [3] **, Noorashikin Md Noor** [1] **, Wan Shafrina Wan Mohd Jaafar** [1] **, Mohd Khairul Amri Kamarudin** [4] **, Effi Helmy Ariffin** [5] **, Nor Aizam Adnan** [6] **and Anizawati Ahmad** [7]

1   Earth Observation Centre, Institute of Climate Change, Universiti Kebangsaan Malaysia, UKM, Bangi 43600, Selangor, Malaysia
2   Department of Civil Engineering, Faculty of Engineering and Built Environment, Universiti Kebangsaan Malaysia, UKM, Bangi 43600, Selangor, Malaysia
3   Department of Surveying Science and Geomatic, Universiti Teknologi MARA, Arau 02600, Perlis, Malaysia
4   East Coast Environmental Research Institute (ESERI), Gong Badak Campus, Universiti Sultan Zainal Abidin, Kuala Nerus 21300, Terengganu, Malaysia
5   Faculty of Science and Marine Environment, Universiti Malaysia Terengganu, Kuala Terengganu 21030, Terengganu, Malaysia
6   Faculty of Architecture, Planning & Surveying, Universiti Teknologi MARA, Shah Alam 40450, Selangor, Malaysia
7   National Water Research Institute of Malaysia (NAHRIM), Lot 5377, Jalan Putra Permai, Seri Kembangan 43300, Selangor, Malaysia
*   Correspondence: knam@ukm.edu.my; Tel.: +60-389-216767

**Abstract:** Coastal areas are fragile and changeable due to natural and anthropogenic factors. The resulting changes could have a significant impact on the coastal community. Thus, monitoring shoreline changes for environmental protection in the Selangor coastal area is an important task to address these issues. The main objective of this study is to analyse the pattern of shoreline changes and predict the shoreline position along the Selangor coast. The geospatial approach can provide information on the history and pattern of shoreline changes. This study used temporal datasets and satellite imagery (SPOT 5) to monitor the shoreline changes throughout the 11 identified study areas. It comprises three methods: shoreline change envelope (SCE), net shoreline movement (NSM), and end-point rate (EPR). The findings indicated that the Selangor coast was more exposed to the erosion phenomenon than to the accretion phenomenon, with 77.3% and 22.7%, respectively. This study reveals significant erosion phenomena in 2 out of 11 areas: Bagan Pasir and Pantai Kelanang. Meanwhile, significant accretion occurred at Bagan Sungai Burong and Sungai Nibong. Consequently, providing complete information would be helpful for researchers, decision-makers, and those in charge of planning and managing the coastal zone.

**Keywords:** DSAS; geospatial; Selangor coast; shoreline changes; coastal erosion

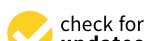



## 1. Introduction

Climate change is a prolonged and complex change in the weather of an area [1]. Naturally, the weather changes every day and even every hour. This phenomenon occurs because of two major factors, which are natural changes and human activity. According to the IPCC, human activity has been a major contributing factor to climate change since the middle of the 20th century [1]. Therefore, natural disasters such as sea-level rise, floods, landslides, coastal erosion, droughts, forest fires, and haze have increased due to the effects of climate change.

Climate change can also be attributed to the increase in global temperature, which is popularly known as global warming [2]. This phenomenon is increasing and is predicted to

continue to rise over time. Increased global sea level is an impact of the melting of glaciers and ice at the poles caused by global warming from human activity. Ice melting in the Arctic is a major factor causing the increase in sea levels and poses a threat, especially to countries that have high rates of population growth and socio-economic activity in the coastal areas. Approximately 400 million people in the world live within 20 kilometres of the coastline [3]. The increase of coastal development is also a major factor that contributed to the boom of the population in the coastal areas. The increase in population in the coastal area is caused by development [4].

Generally, shoreline changes are due to human-related as well as natural phenomena [2]. The phenomenon of shoreline change has an impact not only on communities but also on the economic, physical, and social systems. Multispectral and multitemporal satellite images from low-, medium-, and high-resolution satellites such as SPOT, Landsat, World View, and others can detect shoreline changes. Some researchers used Landsat satellite image analysis (Landsat 7 and Landsat 8) to detect shoreline change while most researchers used the Digital Shoreline Analysis System (DSAS) to analyse the change of shoreline in the selected areas [5]. The erosion along the coastline on the west coast of Peninsular Malaysia is seriously accelerated by human interference in the natural process, such as agriculture and aquaculture activities that involve the construction of tidal gates, bunds, and channel dredging. There have been various types of coastal erosion mitigation plans adopted to mitigate the erosion. The erosion in coastal areas can be controlled either by a soft or hard engineering approach. Common coastal engineering structures applied in Malaysia include breakwater, revetment, groin, seawall, geo-tube, and Simplified Armor Unit-H (SAUH) [5]. Most of these armour units were used on the west coast of Peninsular Malaysia and have been successfully implemented along 1.5 km at Sungai Haji Dorani and Sungai Tegar, Selangor to minimise erosion at the coastline, as shown in Figure 1. A Geo-textile tubes, or geotubes, have been designed as breakwaters for the coasts of Haji Dorani and Kelanang, Selangor [5]. A geotube was positioned just above the lowest tide mark, parallel to the shore. The reason for the installation of this geotube was to form a barrier to break the on-shore waves as well as encourage sedimentation on the shoreward side of the tube [6]. Attempts at the installation of a geo-fabric tube (about two metres in diameter and a hundred metres long) filled with sand, are shown in Figure 2.

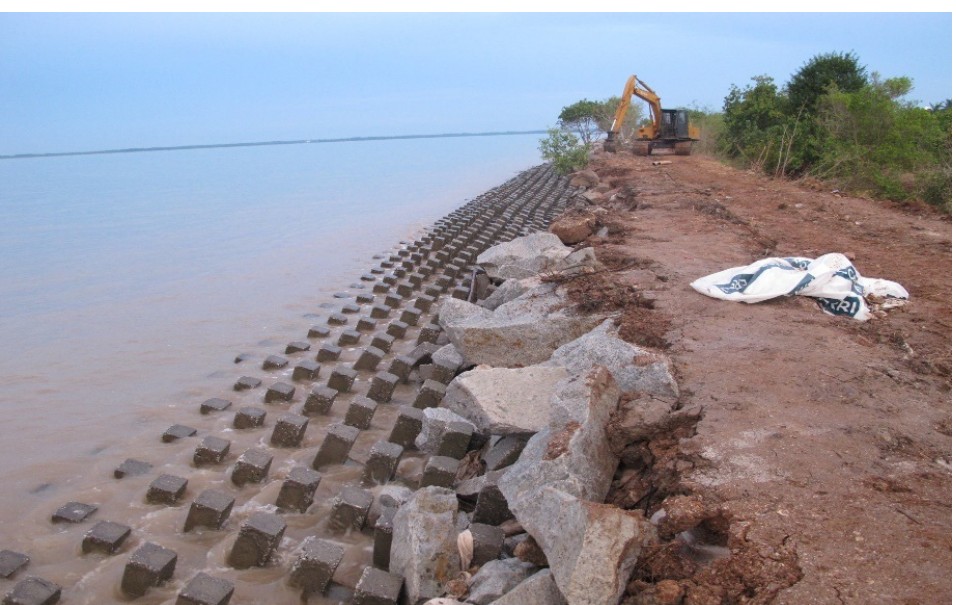

**Figure 1.** Photo showing SAUH installed at Sungai Tegar.

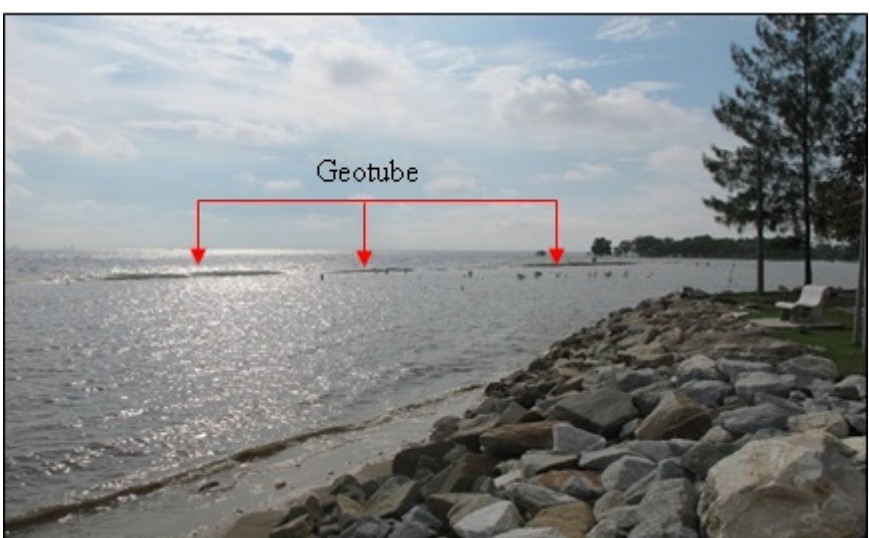

**Figure 2.** Photo showing geotube installed at Pantai Kelanang in 2019.

The coastal zone is the meeting point between the land and the oceans [5]. This zone was categorised as a fragile zone and easily transformed its physical form depending on environmental activity [6]. Furthermore, Mohd et al. [7] also stated in their study that the shoreline changes its shape and size from time to time as a response to the environment. Malaysia is a country that has an 8840-km-long coastline, including Sabah and Sarawak coasts [7], and 1300 km of coastline are facing erosion phenomena [8]. Coastal erosion phenomena are associated with the permanent loss of land and habitat along the shoreline, resulting in coastal transformation [9]. Therefore, monitoring of the coastal zone is an important part of national development and protection of the environment.

Nowadays, improved technologies are necessary for conducting coastal studies. Satellite imagery in conjunction with geographic information systems (GIS) was used to study coastal changes [2,6,10]. In this study, the shoreline changes along the Selangor coast were analysed using a digital shoreline analysis system (DSAS) (version 4.4, Environmental System Research Institute (ESRI), Redlands, CA, USA) [11]. DSAS is a tool to study the shoreline changes using a statistical approach. DSAS yields valuable information about the coastal environmental changes [12]. DSAS has been designed effectively with a friendly user interface, making it easier for users to obtain valuable information for generating an interactive report for analysis [13,14]. This study focuses on identifying the most significant shoreline changes along the Selangor coastal area and predicting future shoreline positions using DSAS analysis.

## 2. Materials and Methods

### 2.1. Study Area

The west coast of Peninsular Malaysia is characterised by mud and a wide coastal area because of its position at the Straits of Malacca, which has moderate wave activity. Selangor has a shoreline that stretches 213 km from Kuala Langat to Sabak Bernam. Almost 71.3% of the shoreline of Selangor has suffered erosion, which involved 151.9 km [13]. Government and private agencies need to take action to address this problem. The study area for this investigation is located on the west coast of Peninsular Malaysia along the Selangor coastal area. Coastal areas have become a major contributor to the economic growth in this area, especially in the tourism and fishing industry sectors. This area was defined as a stretch of land that extended for 1 km from the shoreline. The west coast of Peninsular Malaysia is exposed to the Malacca Straits with a narrow fetch area. The ocean, which is protected by the Sumatra Region, is characterised by mild wave conditions. Most parts of the selected study areas are made of muddy beaches and are rich in diverse species of mangrove forest that promote the growth of marine life and are excellent at dissipating wave energy [10].

Fringe mangrove forests are gradually being washed away in critical areas, such as the Bagan Pasir restoration area to Tanjung Sepat, which has no forest left. The environmental conditions, a description of the existing mangrove stands, and technical details regarding the construction of a hard breakwater (L-block revetment) were observed from Bagan Nakhoda Omar to Sungai Nibong. This study identifies 11 locations from Sabak Bernam to Kuala Langat coastal, namely Bagan Nakhoda Omar, Sungai Pulai, Bagan Sungai Burong, Kampung Haji Dorani, Bagan Pasir, Sungai Nibong, Bagan Sungai Janggut, Pantai Jeram, Pantai Kelanang, Kampung Batu Laut, and Tanjung Sepat (Table 1 and Figure 3).

**Table 1.** List of study areas along Selangor coast.

| No. | Study Area | Latitude | Longitude |
|---|---|---|---|
| 1 | Bagan Nakhoda Omar | 3° 46′ 01.84″ N | 100° 52′ 22.45″ E |
| 2 | Sungai Pulai | 3° 43′ 28.76″ N | 100° 55′ 02.15″ E |
| 3 | Bagan Sungai Burong | 3° 41′ 25.75″ N | 100° 55′ 59.70″ E |
| 4 | Kg Haji Dorani | 3° 38′ 14.30″ N | 101° 01′ 04.99″ E |
| 5 | Sungai Nibong | 3° 35′ 35.47″ N | 101° 03′ 20.59″ E |
| 6 | Bagan Pasir | 3° 23′ 55.91″ N | 101° 10′ 12.14″ E |
| 7 | Pantai Jeram | 3° 13′ 31.25″ N | 101° 18′ 17.26″ E |
| 8 | Bagan Sungai Janggut | 3° 10′ 19.79″ N | 101° 18′ 17.26″ E |
| 9 | Pantai Kelanang | 2° 47′ 18.56″ N | 101° 24′ 43.19″ E |
| 10 | Kg Batu Laut | 2° 43′ 41.16″ N | 101° 27′ 20.45″ E |
| 11 | Tg Sepat | 2° 39′ 27.87″ N | 101° 33′ 19.05″ E |

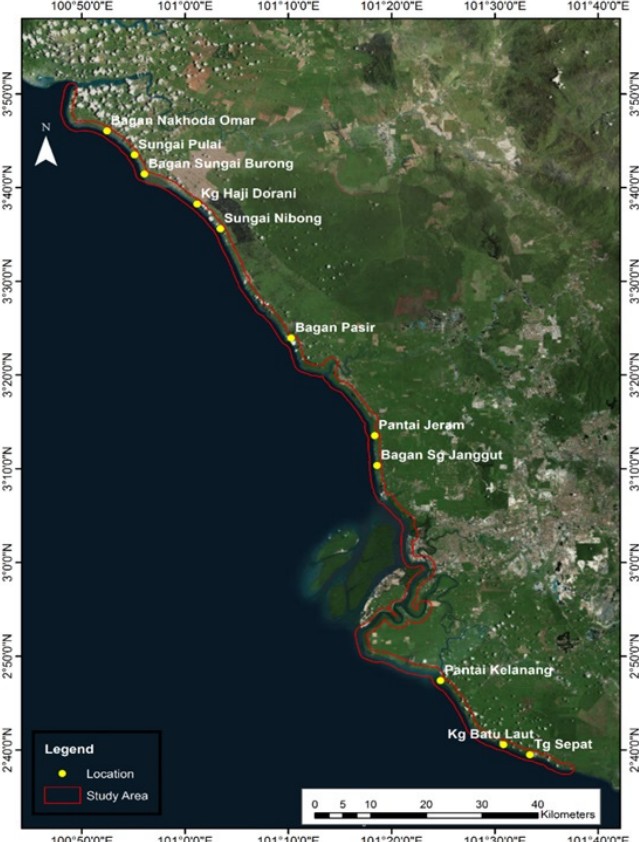

**Figure 3.** Study areas along coast of Selangor represented by the red line.

### 2.2. Data Sources

The data sources used in this study were acquired at different temporal resolutions. Temporal resolution denotes the data acquisition between 1993 and 2014 to investigate the shoreline changes. Table 2 shows the data sources according to the type of data, data

acquisition, spatial resolution, and scale. The projection system of a satellite image that was used in this study is the Rectified Skew Orthomorphic (RSO) projection system. This study was carried out to investigate the shoreline changes from the years 1993 to 2014 using topographic data and SPOT 5 satellite imagery. SPOT-5 images have four multi-spectral bands with a spatial resolution of 10 m in the green (500–590 nm), red (610–680 nm), near-infrared (NIR, 780–890 nm), and short-wavelength infrared (SWIR, 1580–1750 nm), also, it has one panchromatic band (480–710 nm) with a spatial resolution of 2.5 m. These images are suitable for shoreline extraction as SPOT-5 includes two of the spectral bands that are most effective for visualization of the shoreline with the combination of MIR, NIR, and green bands [7].

**Table 2.** Data sources according to the type of data, data acquisition, and spatial resolution/scale.

| Type of Data | Date of Acquisition | Spatial Resolution/Scale |
|---|---|---|
| Topographic | 1993 | 1: 50 000 |
| | 14/12/2004 | 2.5 m |
| | 03/04/2005 | 2.5 m |
| | 10/09/2006 | 2.5 m |
| | 07/06/2007 | 2.5 m |
| SPOT 5 | 13/11/2005 | 2.5 m |
| | 25/04/2013 | 2.5 m |
| | 01/09/2014 | 2.5 m |
| | 04/05/2014 | 2.5 m |

### 2.3. Shoreline Analysis

There are a few steps needed to compute the analysis, including: (1) historical shoreline position preparation, (2) baseline determination, (3) transect generation, and (4) change computation. The shoreline datasets for the years 1993, 2004, and 2014 were extracted through a digitising technique using ArcGIS software. In this study, the baseline is set as a starting reference point for the transects to compute the distance changes. The buffer technique was used in this study to determine the baseline. The baselines were drawn at 1 km, parallel to the shoreline position. The baseline was drawn parallel to shoreline in 1993 based on the vegetation line. The aerial images were auto-mosaiced based on ground control point (GCP) referencing to fix permanent landmarks [11]. The analysis was performed at the same time for each study site. Meanwhile, the transect line was automatically generated by the software and this study generated 2321 transects with a 2.5 m interval between each transect. Every transects has its own unique numbers. The DSAS executes five statistical operations, but in this study, only three statistical operations were used, namely, shoreline change envelop (SCE), net shoreline movement (NSM), and endpoint rate (EPR). The benefit of using these statistical operations is the ability to compute the rate-of-change statistics for a time series of shoreline positions. The statistics allow the nature of shoreline dynamics and trends in change to be evaluated and addressed. The SCE method calculates the distance of shoreline between the farthest and closest points from the baseline without taking their shoreline dates. Equation below was used to calculate SCE [12]:

$$S_d = f_x - f_y$$

where $S_d$ represents the shoreline distance (m); $f_x$ is the distance of the further shoreline from the baseline (m); and $f_y$ is the distance of the shoreline closest to the baseline (m).

The NSM method was used to calculate the distance between the oldest and most recent shorelines for each transect using only two historical shoreline positions. The formula for NSM is illustrated using equation as follows [12]:

$$S_m = f_n - f_m$$

where $S_m$ denotes the net of shoreline movement (m); $f_n$ denotes the distance between baseline and shoreline for the oldest shoreline (m); and $f_m$ denotes the distance between baseline and shoreline for the most recent shoreline position along the same transects. In fact, both NSM and SCE do not report the rate value, but these methods represent the shoreline changes in distance value.

The EPR method was computed by dividing the total distance of shoreline changes by the time period. Generally, this method calculates the annual rate of change. The EPR result illustrated the trend of erosion and accretion that occur in the coastal zone. The formula for EPR is shown in equation [13], as follows:

$$\text{EPR} = \frac{\text{distance A} - \text{B}}{\text{time between youngest and oldest shoreline}}$$

where EPR is the rate, and distance A and B are the distance of youngest and oldest shoreline from baseline in the unit of meters.

*2.4. Shoreline Changes Prediction*

The prediction of shoreline position in the future is an important element in facilitating the process of coastal management. The model makes a prediction based on the assumed rate of shoreline change and a good estimation method for predicting future shoreline position [14]. In this study, EPR results were implemented to predict the shoreline position, as the EPR result is easier and more practical for making a future shoreline prediction [15]. This model used slope (rate) and y-intercept to estimate the shoreline position. Other than that, additional information that is related to the shoreline study, such as wave, wind, current, tide, and sediment transport, is not required in the prediction process [16]. The equation for shoreline prediction position is shown, as follows [17]:

$$\text{shoreline position} = \text{rate per year} * \text{time period} + \text{y-intercept}$$

The calculation of the EPR model is based on equation [18] below which uses two historical shoreline positions, where the earliest position is denoted as $S_1$ and the latest position is denoted as $S_2$. SP was used to denote the shoreline positions, $T$ for time (date interval), $M_{EPR}$ for the rate of shoreline changes, and $B_{EPR}$ for model intercept.

$$SP = M_{EPR} * T + B_{EPR}$$

Equation rate of shoreline ($M_{EPR}$), calculated as:

$$M_{EPR} = (S_2 - S_1)/(T_2 - T_1)$$

Equation EPR intercept, calculated as:

$$B_{EPR} = S_1 - (M_{EPR} * T_1) = S_2 - (M_{EPR} * T_2)$$

Since the endpoint of the line can extend beyond the most recent point, (P) can be rewritten to use that position ($S_2$) and the elapsed time ($Tp - T_2$)

$$P = M_{EPR} * (Tp - T_2) + S_2$$

## 3. Results and Discussion

The results of shoreline change were obtained using DSAS analysis, for which the study used SCE, NSM, and EPR methods. SCE results highlighted the distance of shoreline between the furthest and closest from baseline for transects (Table 3). These results represent the total of distance change in shoreline movement for all shoreline datasets regardless of the dates. Based on the result, Bagan Sungai Janggut was determined as the minimum SCE value, at 0.89 m. Meanwhile, Bagan Sungai Burong recorded the highest SCE value,

at 1527.0 m. This was influenced by the natural activities occurring along the shoreline, where Bagan Sungai Burong was recognized as a mangrove area. In addition, some of these study areas (e.g., Bagan Sungai Burong and Sungai Nibong) have undergone beach nourishment processes that are indirectly helping to improve the accretion phenomena in these areas. Overall, the patterns of change shown indicate a dominance of erosion occurring at unequal levels.

**Table 3.** Minimum and maximum distance of shoreline from baseline for the transects.

| Study Area | Min (m) | Max (m) |
|---|---|---|
| Bagan Nakhoda Omar | 8.47 | 429.50 |
| Sungai Pulai | 103.40 | 317.39 |
| Bagan Sungai Burong | 14.24 | 1527.00 |
| Kg Haji Dorani | 7.10 | 170.07 |
| Sungai Nibong | 12.22 | 180.43 |
| Bagan Pasir | 62.88 | 634.17 |
| Pantai Jeram | 1.12 | 211.19 |
| Bagan Sungai Janggut | 0.89 | 179.12 |
| Pantai Kelanang | 0.93 | 280.30 |
| Kg Batu Laut | 19.57 | 242.03 |
| Tg Sepat | 2.53 | 325.30 |

The result of the NSM calculation is not a rate value but reported as a distance change between the oldest and the most recent shoreline. Shoreline change was measured between the years 1993 and 2014 (Table 4), as previously stated in Equation (2) of the methods section. Bagan Pasir has indicated the maximum erosion change, at 634.17 m, while the minimum erosion change occurred in Sungai Nibong, at 0.12 m. On the other hand, Bagan Sungai Burong has experienced the maximum erosion change, at 1527.06 m, while Pantai Jeram was found to be the area with the minimum accreting phenomena. The findings by Selamat et al. [10] showed that a serious erosion problem occurred in the Bagan Pasir area. The findings were consistent with the shoreline erosion findings by Asmawi and Ibrahim [19], which showed that most of the shoreline in Kuala Selangor was suffering from coastal erosion that was classified as extremely dangerous. The results of this study also provide stronger supporting evidence about the erosion problem of the Selangor coastline, as pointed out in a previous study by Daud et al. [5], whereby the average erosion rates along the entire Selangor coastline were more than 2 m/year. Erosion along the coastline on the west coast of Peninsular Malaysia is greatly accelerated by human interference in the natural process, such as agriculture and aquaculture activities that involve the construction of tidal gates, bunds, and channel dredging [20].

**Table 4.** Result of the Net Shoreline Movement (NSM) along the study areas.

| Study Area | Erosion (m) | | | Accretion | | |
|---|---|---|---|---|---|---|
| | Max | Min | % | Max | Min | (%) |
| Bagan Nakhoda Omar | 246.04 | 4.21 | 76.2 | 343.92 | 1.33 | 23.8 |
| Sungai Pulai | 317.39 | 103.40 | 100.0 | 0 | 0 | 0.0 |
| Bagan Sungai Burong | 471.26 | 43.65 | 38.5 | 1527.06 | 14.24 | 61.5 |
| Kg Haji Dorani | 170.07 | 4.32 | 97.6 | 41.34 | 9.15 | 2.4 |
| Sungai Nibong | 113.35 | 0.12 | 38.5 | 164.84 | 0.40 | 61.5 |
| Bagan Pasir | 634.17 | 27.53 | 98.4 | 91.66 | 9.22 | 1.6 |
| Pantai Jeram | 145.91 | 0.20 | 79.7 | 211.19 | 0.38 | 20.3 |
| Bagan Sungai Janggut | 179.12 | 0.89 | 90.7 | 85.05 | 2.49 | 9.3 |
| Pantai Kelanang | 258.32 | 0.93 | 72.2 | 142.54 | 0.89 | 27.8 |
| Kg Batu Laut | 242.03 | 0.40 | 68.4 | 97.84 | 1.16 | 31.6 |
| Tg Sepat | 325.30 | 1.14 | 89.7 | 168.70 | 3.08 | 10.3 |

The percentage of erosion and accretion among 11 study locations between 1993 and 2014 indicates that 77.3% of the study area is facing erosion, while only 22.7% of the study area is facing accretion (Figure 4). In addition, this study finds that Sungai Pulai has experienced a 100% erosion phenomenon from 1993 to 2014. This is because Sungai Pulai is located on a low-lying land area. Garbage on land proves that water is over the shoreline area whenever there is a high-tide event. (Figure 5). Sea level rise has a major effect on human activities and the physical environment along the Selangor coastal area because part of the shoreline is a low-lying area [21]. The percentage of erosion at Bagan Pasir is 98.4%. This area has suffered erosion phenomena because they do not have any mitigation or hard structure to prevent erosion. Meanwhile, Kg Haji Dorani was recorded as experiencing the third-highest percentage of erosion, at 97.6%. The categories of land use activity in the coastal zone will affect the phenomena of shoreline changes. There are several activities that affect erosion at Kg Haji Dorani, such as the development of the resort and aquaculture. Although Kg Haji Dorani has mangroves, they do not face the tide and waves in this area. Half of the erosion phenomena in Malaysia occur in muddy and mangrove- fringed areas [22]. The high percentage of accretion phenomena occurred at Bagan Sungai Burong and Sungai Nibong. The phenomenon of accretion in these areas is due to mangrove activities. As a result, mangroves were identified as an alternative to reduce erosion.

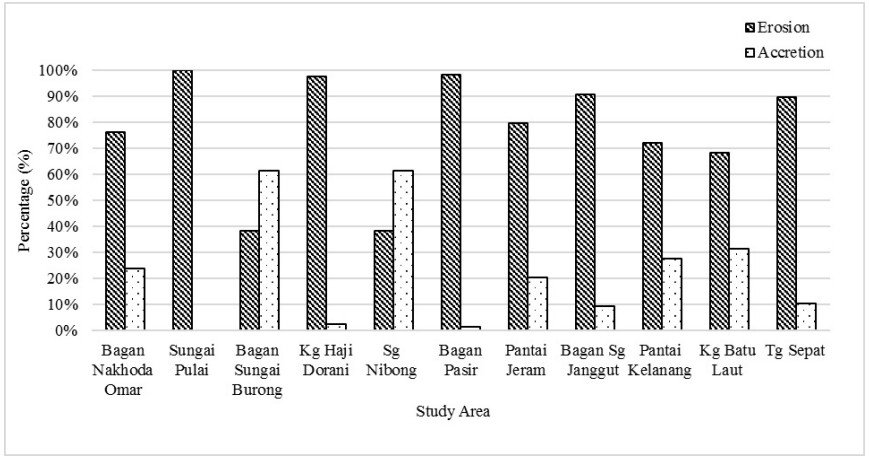

**Figure 4.** The variation of the erosion and accretion phenomena along the study area.

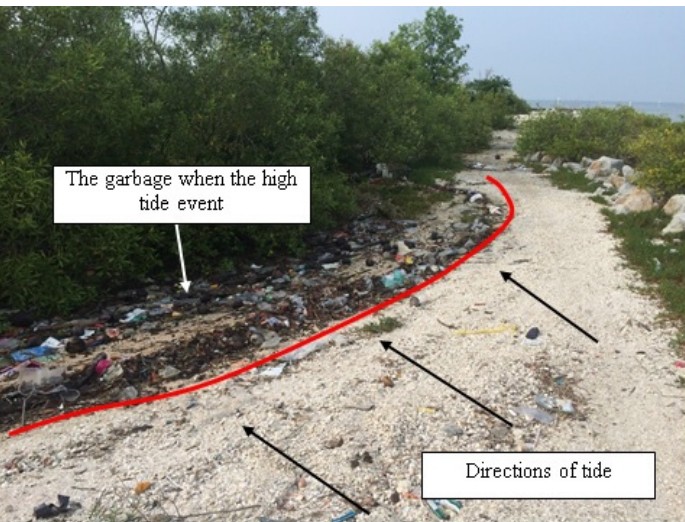

**Figure 5.** Photo showing the accumulation of garbage as the high tide occurred at Sungai Pulai (2017). Red line indicating erosion along the shoreline.

The EPR result is calculated by dividing the distance of shoreline by the time between the oldest and the most recent shoreline [11]. The positive rate of shoreline changes indicates accretion and the negative value shows erosion (Table 5), computed from Equation (3), as previously stated in the methods section. This study revealed that during the years 1993 to 2004, Sungai Pulai and Bagan Pasir experienced the minimum mean rate of change, with −16.15 m and −12.36 m, respectively, while the maximum mean rate of shoreline was located at Bagan Sungai Burong. The study also showed the rate of shoreline changes from the years 2004 to 2014. Based on the results, the minimum mean shoreline change that occurred at Bagan Sungai Janggut was −11.69 m. Meanwhile, the maximum mean of shoreline changes was obtained in Bagan Sugai Burong. Shoreline change statistics (NSM and EPR) presented for this study demonstrated the large-scale patterns of growth of the affected shorelines [23]. Within our spatial resolution, we classified eight types of shorelines: very high erosion, high erosion, moderate erosion, stable, moderate accretion, high accretion, and very high accretion. As observed in Figure 6, there was a clear tendency of erosion at the shoreline in 1993 until the most retreated position in 2015 at Bagan Pasir and Pantai Kelanang. Meanwhile, Bagan Sungai Burong and Sungai Nibong showed a pattern of high accretion within the 10-year timeframe of the study.

**Table 5.** Rate of shoreline changes using the endpoint rate (EPR) method.

| | 1993–2004 | | | 2004–2014 | | |
|---|---|---|---|---|---|---|
| **Study Area** | **Min** | **Max** | **Mean** | **Min** | **Max** | **Mean** |
| Bagan Nakhoda Omar | −20.93 | 0.52 | −8.24 | −3.31 | 45.77 | 6.45 |
| Sungai Pulai | −25.16 | −7.59 | −16.15 | −142.67 | 1.96 | −0.69 |
| Bagan Sungai Burong | −36.89 | 84.39 | 35.62 | −33.32 | 61.53 | 10.46 |
| Kg Haji Dorani | −10.79 | 3.37 | −4.78 | −7.50 | 3.55 | −2.89 |
| Sungai Nibong | −8.69 | 9.17 | −3.14 | −5.72 | 19.86 | 6.37 |
| Bagan Pasir | −34.26 | 13.82 | −12.37 | −35.95 | 19.32 | −7.70 |
| Pantai Jeram | −0.34 | 4.94 | 0.16 | −21.80 | 28.03 | −4.21 |
| Bagan Sungai Janggut | −0.35 | 8.33 | 0.10 | −26.92 | 7.06 | −11.69 |
| Pantai Kelanang | −19.43 | 9.88 | −4.32 | −36.96 | 32.45 | −4.72 |
| Kg Batu Laut | −18.72 | 14.69 | 1.65 | −31.38 | 24.92 | −7.58 |
| Tg Sepat | −20.48 | 6.21 | −6.79 | −21.44 | 20.49 | −4.81 |

The prediction of the future shoreline of this study has been calculated by using the EPR model for the years 2030 and 2040. The validation of the EPR model was run by comparing the positional difference to the extracted shoreline of 2014. Models based on EPR have been widely used in the predication of future temporal changes of the shoreline [24]. Those models assume that the historically observed change rates of the shoreline can capture the cumulative effects of coastal processes, such as waves, currents, and storms [25]. The basic assumption of the EPR model is that the observed historical rates of the shoreline changes are the best estimates for predicting its future position. The maximum accretion changes occurred in Bagan Sungai Burong and the maximum erosion changes occurred in Pantai Kelanang and Bagan Pasir. Additionally, based on the shoreline analysis of the years 2030 and 2040, Pantai Kelanang and Bagan Pasir were dominated by erosion, while accretion was predicted for Bagan Sungai Burong area. These findings are supported by a study conducted by Zecchin et al. [26], which revealed that the presence of the large grain-size shell hash along the coast indicates that these locations are still experiencing some major wave and current erosion. Based on the predictions for the years 2030 and 2040, no significant changes will have occurred at Sungai Pulai (Figures 7 and 8).

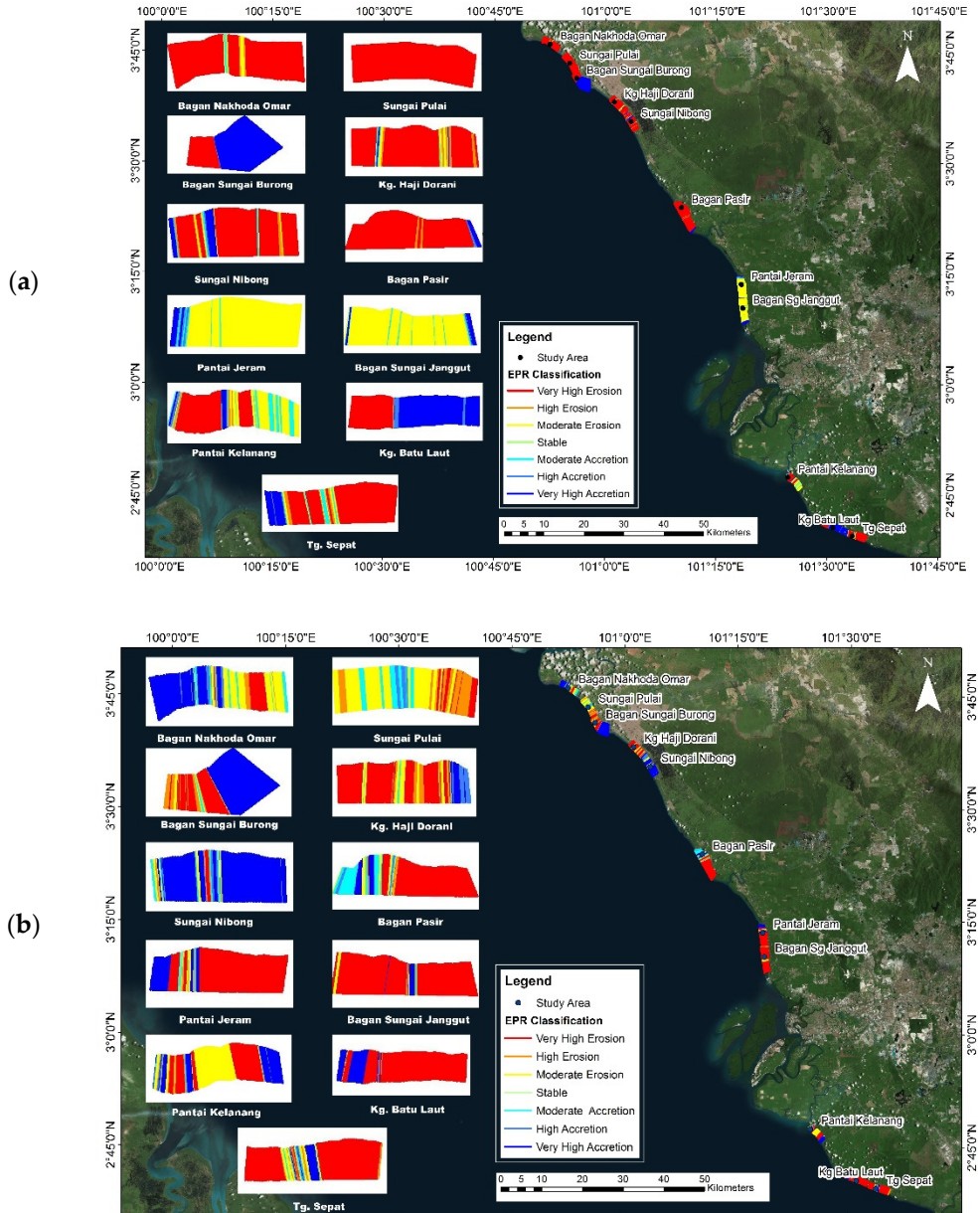

**Figure 6.** Shoreline change along the study area from (**a**) 1993–2004 and (**b**) 2004–2014.

In addition, the lower rate of shoreline changes throughout 2004 until 2014 was a major factor that caused Sungai Pulai to not significantly change. This study supports a piece of evidence from the previous study, which found that Selangor has been classified as having the most serious coastal area issue. Other than that, the majority of the community in Selangor stated that the erosion phenomenon has become the most serious and dangerous issue for the communities along the Selangor coast [27]. Hence, this is evidence of the impact of the shoreline change causes of land loss that occurred in Bagan Pasir and facilities damage in Pantai Kelanang, based on monitoring and observation (Figures 9 and 10).

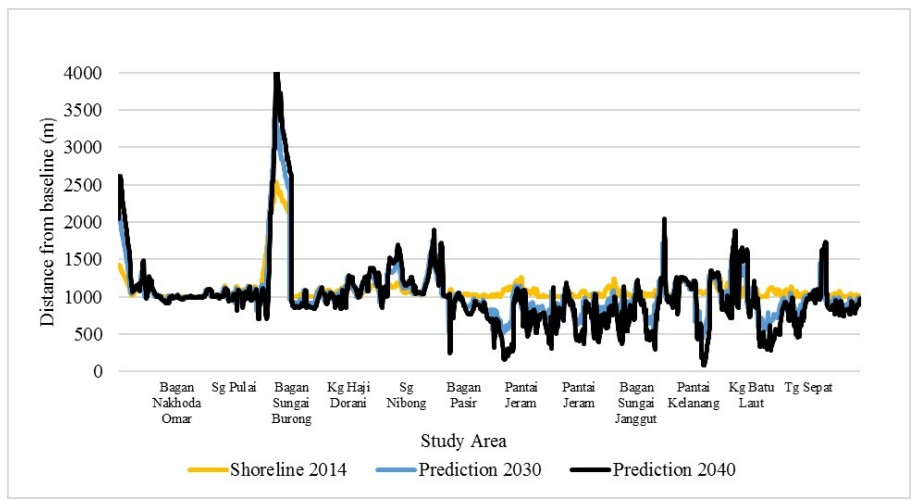

**Figure 7.** Prediction of shoreline changes for the years 2030 and 2040.

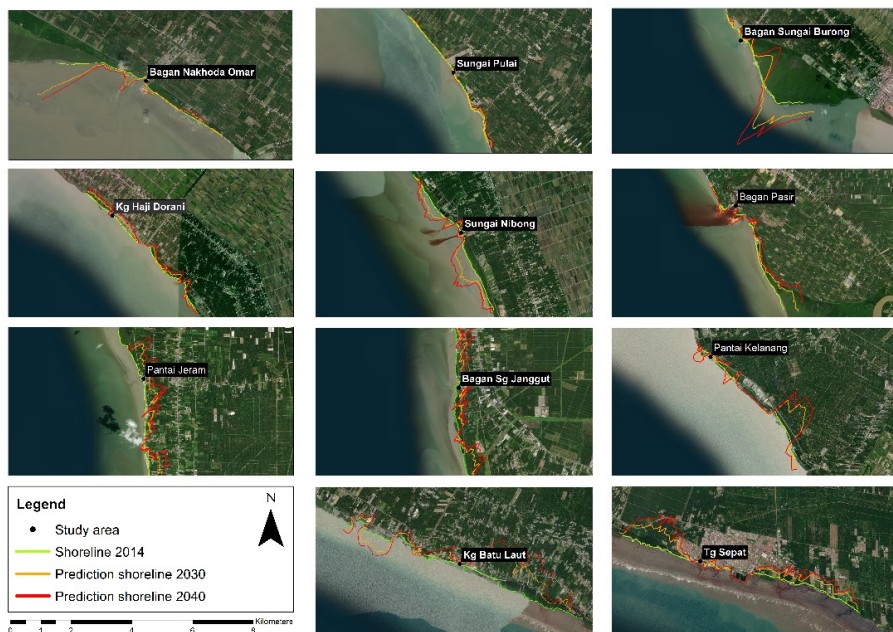

**Figure 8.** Predicted shoreline changes in 2030 (orange line) and 2040 (red line) based on 2014 shoreline (green line).

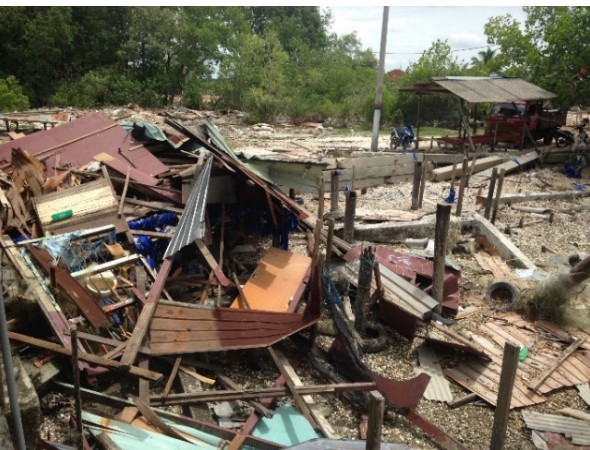

**Figure 9.** Land loss impacts on erosion phenomena in Bagan Pasir.

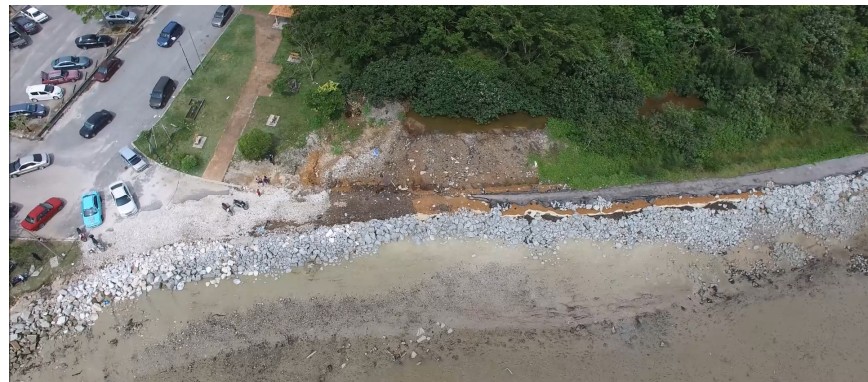

**Figure 10.** Facilities damage impacts on shoreline changes in Pantai Kelanang.

The shoreline change statistics (SCE, NSM, and EPR) reported for the case study were able to demonstrate the large-scale patterns of retreat and expansion of the shorelines in the research study. Different statistical methods were similarly successful at revealing the spatial and temporal movement patterns [28]. SCE was chosen due to its ability to provide the variation envelope and because it is the sole metric that takes into consideration all shorelines. The NSM and EPR only show the difference between the initial and most recent measurements. The selection of DSAS statistical parameters in the case study enabled the investigation of the temporal and spatial dynamics of the coastal change and the geomorphic variability along the coastline due to their capacity to utilise all coastline positions (SCE), the cumulative shoreline movement (NSM), and the time variations (EPR) that encompass the rate range of the set of data [29]. The SCE measures the total change in coastline, considering all available coastline positions, and reports on their distance without reference to a specific date, and the NSM measures the distance of coastline change between the longest coastline and the newest coastline, while the EPR calculates the speed of shoreline change by dividing the distance between the longest coastline and the most recent coastline by time. EPR is suggested to be the most applicable in this study for measuring the speed of shoreline changes [30]. This is supported by a previous study, which described that the advantage of this algorithm is that it requires only two shoreline dates for its computation; its disadvantage appears when there are many shorelines, as all intermediate shorelines dates are ignored [31].

**4. Conclusions**

Based on the results of this study, it can be said that combining satellite imagery with a GIS approach can give accurate and useful information about how shorelines change over time. Between 1993 and 2014, three-quarters of the study area was subjected to erosion phenomena, while only one-quarter was subjected to accretion phenomena. Bagan Pasir and Pantai Kelanang were highlighted as the most significantly eroded areas. Meanwhile, Bagan Sungai Burong and Sungai Nibong have been analysed as the most significant accretion areas, with the same percentage of accretion. Although mitigation has been done in the Selangor coastal area, as it helps in reducing extreme wave and tidal impacts, some of the affected locations are unable to reduce these impacts. In further studies, we expect to conduct a comprehensive analysis that will consider seasonal behaviour to improve the understanding of coastal change. Generally, the Selangor coastal area is more prone to erosion than accretion phenomena. Any coastal activities and development will have an indirect impact on shoreline changes, and at the same time, the adverse effects will be faced by the coastal community in the future. Therefore, information and history about how the shoreline has changed are important for improving how well coastal management works.

**Author Contributions:** Conceptualization, K.N.A.M., S.N.S., F.A.M., W.S.W.M.J., M.K.A.K., E.H.A., N.M.N., N.A.A. and A.A.; methodology, K.N.A.M. and S.N.S.; software, K.N.A.M., S.N.S. and F.A.M.; validation, K.N.A.M., S.N.S. and F.A.M.; formal analysis, K.N.A.M., S.N.S., F.A.M., W.S.W.M.J.,

M.K.A.K. and E.H.A.; investigation, K.N.A.M. and S.N.S.; resources, K.N.A.M. and S.N.S.; data curation, K.N.A.M., S.N.S., F.A.M., W.S.W.M.J., M.K.A.K. and E.H.A.; writing—original draft preparation, K.N.A.M., S.N.S.; writing—review and editing, K.N.A.M., S.N.S.; F.A.M., W.S.W.M.J., M.K.A.K. and N.M.N.; visualization, K.N.A.M. and S.N.S.; supervision, K.N.A.M.; project administration, K.N.A.M.; funding acquisition, K.N.A.M. All authors have read and agreed to the published version of the manuscript.

**Funding:** This research was funded by UKM YSD Chair of Sustainability (UKM-YSD-2021-003) and the Trans-Disciplinary Research Grant Scheme (TRGS/1/2015/UKM/02/5/1).

**Institutional Review Board Statement:** Not applicable.

**Informed Consent Statement:** Not applicable.

**Data Availability Statement:** Not applicable.

**Acknowledgments:** The authors gratefully acknowledge the Earth Observation Centre, Institute of Climate Change, Universiti Kebangsaan Malaysia (UKM) for sharing the data to conduct this investigation. This study was supported by the Research Fund provided by the research grants of the UKM YSD Chair of Sustainability (UKM-YSD-2021-003) and the Trans-Disciplinary Research Grant Scheme (TRGS/1/2015/UKM/02/5/1).

**Conflicts of Interest:** The authors declare no conflict of interest.

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
