# Peer review of "Assessment of Shoreline Changes for the Selangor Coast, Malaysia, Using the Digital Shoreline Analysis System Technique"

_urbansci, doi:10.3390/urbansci6040071_

Round 1

Reviewer 1 Report

The authors present the assessment of shoreline changes along the coast of Selangor in Malaysia. The multi-temporal remote sensing imageries are utilized together with the DSAS system in the ESRI software. Multi-temporal SPOT-5 imageries from 2004-2005 and 2013-2014 are digitized and then processed using DSAS for shoreline analysis. There are 3 main statistical operations are concerned: SCE, NSM and EPR. Finally, with the EPR, the future position (2030 and 2040) of the shoreline is predicted over 11 areas.

The manuscript is generally well-written except for minor grammatical errors, thanks to the author for their efforts.

The findings reported in this paper are indeed important for coastal management particularly when it provides the rate of erosion and accretion for specific (11) study sites around Selangor. Although the manuscript has an important message, I found the manuscript is lack the necessary detail for readers to fully understand the techniques and the interpretation of the results/analysis. Further scientific evidence is required to support the analysis and the conclusion. There are several issues that need to be clarified.

Therefore, I encourage the authors to consider my comments below.

General comments

1)     The paper has provided an analysis of the shoreline changes from DSAS. However, it is missing the results (output from the imageries) and the validation of the extracted shorelines. This is the major shortcoming of the paper. While it is claimed that the geospatial approach is used, there are no geospatial-based results being presented to demonstrate the shoreline changes in spatial form. Please consider adding the results of Spot-5 shoreline changes and the spatially predicted shorelines for both 2030 and 2040. In addition, the results should be validated. This is somehow crucial particularly when the analysis is obtained from the eye-of-the-sky (satellite) datasets. Validation should give an overview of how accurate and reliable are the study results.

Specific comments

1)     Line 27-28: “the main objective…to analyse the pattern of shoreline changes and predict the shoreline position.” I doubt the results and analysis presented in this paper really give the answer to these objectives since the analysis and conclusions have never discussed the pattern of shoreline changes. Similar concerns are also for “to predict the shoreline position”, when there is no figure/map to show the position of the future shoreline.

2)     Line 28: “geospatial approach”, but no geospatial-based results are being presented.

3)     Section 1: Could authors please add more literature reviews related to the work in Malaysia? The motivation of the study is missing from the introduction. Why the such study is important?

4)     Table 2: table caption and table heading can be better written. E.g., Parameters==date of acquisition. Capabilities values ==scales/spatial resolutions. Please check if the information in the “Parameters” column is correct (12/11/2005, and repeated 2013).

5)     Line 212-122: The shoreline changes are assessed using three statistical methods. Could author please expand the choice? Why are these three important?

6)     Section 3: I recommend authors show the spatial map of the shoreline changes from the three methods, and also the future predicted shoreline.

7)     It is difficult to understand the difference between the three methods. Line  179 states SCE is the distance of shoreline between the further and closest baseline…Line 189 states the NSM is the distance change between the oldest and the most recent shoreline…Line 238 states the EPR is the distance shoreline by the time between the oldest and the most recent shoreline. Is this imply that the unit for the NSM and EPR is m/yr? Because it seems they both are relatives to the time.

8)     Table 3, 4 and 5. Results from this table are under-discussed. The author’s discussion is merely among the min and max changes. How about their pattern? (as this is one of the objective).

9)     Line 193: “…Bagan Sg. Burong indicated the maximum erosion…” please check if this is correct.

10)  Results from Table 3 and 4 should be integrated. It seems the results in Table 3 (e.g. Sg.Pulai, Bagan Sg.Burong, Pantai Jeram and Bagan Sg. Janggut) and some other values are similar to those found in Table 4. Please discuss this matter. Also, I wonder why we need results from Table 3 when it only can give us the information about min/max, while Table 4 can give more comprehensive results (accretion/erosion). Can we just exclude Table 3 from the paper? Please justify.

11)  Table 5: Why the analysis is divided into 2 different time frames? Why such analysis is not important for Tables 3 and 4? Again, the results from this table are under-discussed.

12)  After the discussion in Section 3, I wonder which methods are the best and recommended for future studies? Proper evidence should be provided.

13)  Line 348-350: “…the most significantly eroded areas” and “…the most signifanct accretion”. I presume, this erosion category should be established (probably in the methodology) prior to the conclusions. Another thing, “Sg. Burong is categorised as the most significant accretion area”, but it seems Sg. Nibong is significantly eroded provided that they have the percentage >60%. Similar concerns to “the most significant erosion”. 

Author Response

Paper Urban Science 1857789

We appreciate the comments from the respected reviewers. Here is the list of comments from the reviewers with our responses:

Title: Assessment Of Shoreline Changes For The Se-Langor Coast, Malaysia, Using The Digital Shoreline Analysis System Technique

Reviewer 1

The authors present the assessment of shoreline changes along the coast of Selangor in Malaysia. The multi-temporal remote sensing imageries are utilized together with the DSAS system in the ESRI software. Multi-temporal SPOT-5 imageries from 2004-2005 and 2013-2014 are digitized and then processed using DSAS for shoreline analysis. There are 3 main statistical operations are concerned: SCE, NSM and EPR. Finally, with the EPR, the future position (2030 and 2040) of the shoreline is predicted over 11 areas.

The manuscript is generally well-written except for minor grammatical errors, thanks to the author for their efforts.

The findings reported in this paper are indeed important for coastal management particularly when it provides the rate of erosion and accretion for specific (11) study sites around Selangor. Although the manuscript has an important message, I found the manuscript is lack the necessary detail for readers to fully understand the techniques and the interpretation of the results/analysis. Further scientific evidence is required to support the analysis and the conclusion. There are several issues that need to be clarified.

Therefore, I encourage the authors to consider my comments below.

Response: Thank you for the positive comments. We have considered the comments suggested by the reviewer as stated below.

General comments

Point 1: The paper has provided an analysis of the shoreline changes from DSAS. However, it is missing the results (output from the imageries) and the validation of the extracted shorelines. This is the major shortcoming of the paper. While it is claimed that the geospatial approach is used, there are no geospatial-based results being presented to demonstrate the shoreline changes in spatial form. Please consider adding the results of Spot-5 shoreline changes and the spatially predicted shorelines for both 2030 and 2040. In addition, the results should be validated. This is somehow crucial particularly when the analysis is obtained from the eye-of-the-sky (satellite) datasets. Validation should give an overview of how accurate and reliable are the study results.

Response 1: The results of SPOT-5 shoreline changes has been added and discussed in the results and discussion accordingly. Please see line 302-306, 337-339, 347-366.

Specific comments

Point 1:  Line 27-28: “the main objective…to analyse the pattern of shoreline changes and predict the shoreline position.” I doubt the results and analysis presented in this paper really give the answer to these objectives since the analysis and conclusions have never discussed the pattern of shoreline changes. Similar concerns are also for “to predict the shoreline position”, when there is no figure/map to show the position of the future shoreline.

Response 1: Related figures has been added in the results based on SPOT-5. See line 302-306, 337-339.

Point 2:  Line 28: “geospatial approach”, but no geospatial-based results are being presented.

Response 2: Figures based on SPOT-5 results has been presented in the result. See line 302-306, 337-339.

Point 3: Section 1: Could authors please add more literature reviews related to the work in Malaysia? The motivation of the study is missing from the introduction. Why the such study is important?

Response 3: Literature reviews has been added into introduction part on study importance. Please see line 58-96.

Point 4: Table 2: table caption and table heading can be better written. E.g., Parameters==date of acquisition. Capabilities values ==scales/spatial resolutions. Please check if the information in the “Parameters” column is correct (12/11/2005, and repeated 2013).

Response 4: Table caption and heading has been edited with the information has been checked. See line 155.

Point 5: Line 212-122: The shoreline changes are assessed using three statistical methods. Could author please expand the choice? Why are these three important?

Response 5: Justification on the statistical method has been added in the paragraph. Please see line 169-172, 347-366.

Point 6: Section 3: I recommend authors show the spatial map of the shoreline changes from the three methods, and also the future predicted shoreline.

Response 6: The spatial map of the shoreline changes with future predicted shoreline has been added. See line 302-306, 337-339.

Point 7: It is difficult to understand the difference between the three methods. Line  179 states SCE is the distance of shoreline between the further and closest baseline…Line 189 states the NSM is the distance change between the oldest and the most recent shoreline…Line 238 states the EPR is the distance shoreline by the time between the oldest and the most recent shoreline. Is this imply that the unit for the NSM and EPR is m/yr? Because it seems they both are relatives to the time.

Response 7: The SCE measures the total change in coastline considering all available coastline positions and reports on their distance, without reference to a specific date, and the NSM measures the distance of coastline change between the longest coastline and the newest coastline, while the EPR calculates the speed of shoreline change by dividing the distance between the longest coastline and the most recent coastline by time. EPR is suggested to be the most suitable method for measuring the speed of shoreline changes. This is supported by previous study which described that the advantage of this algorithm is that it requires only two shoreline dates for its computation, its inconvenience appears when there are many shorelines, all intermediate shorelines dates are ignored. See line 347-366.

Point 8: Table 3, 4 and 5. Results from this table are under-discussed. The author’s discussion is merely among the min and max changes. How about their pattern? (as this is one of the objective).

Response 8: The shoreline changes pattern has been added in the results paragraph as highlighted in line 232-234, 241-244, 284-292.

Point 9: Line 193: “…Bagan Sg. Burong indicated the maximum erosion…” please check if this is correct.

Response 9: Sentence has been corrected accordingly. See line 243.

Point 10: Results from Table 3 and 4 should be integrated. It seems the results in Table 3 (e.g. Sg.Pulai, Bagan Sg.Burong, Pantai Jeram and Bagan Sg. Janggut) and some other values are similar to those found in Table 4. Please discuss this matter. Also, I wonder why we need results from Table 3 when it only can give us the information about min/max, while Table 4 can give more comprehensive results (accretion/erosion). Can we just exclude Table 3 from the paper? Please justify.

Response 10: Table 3 was computed from SCE method while Table 4 was computed from NSM method. The SCE measures the total change in coastline considering all available coastline positions and reports on their distance, without reference to a specific date, and the NSM measures the distance of coastline change between the longest coastline and the newest coastline. Thus, these two tables are needed to present the values derived from the two different methods.

Point 11: Table 5: Why the analysis is divided into 2 different time frames? Why such analysis is not important for Tables 3 and 4? Again, the results from this table are under-discussed.

Response 11: In Table 5, EPR is calculated by dividing the distance between the longest coastline and the most recent coastline by time. While Table 4 (NSM) measures the distance of coastline change between the longest coastline and the newest coastline, thus time is not considered in the table 4 as table 5.

Point 12: After the discussion in Section 3, I wonder which methods are the best and recommended for future studies? Proper evidence should be provided.

Response 12: EPR is suggested to be the most suitable method for measuring the speed of shoreline changes. Justification has been added in discussion. Please see line 362-366.

Point 13:  Line 348-350: “…the most significantly eroded areas” and “…the most signifanct accretion”. I presume, this erosion category should be established (probably in the methodology) prior to the conclusions. Another thing, “Sg. Burong is categorised as the most significant accretion area”, but it seems Sg. Nibong is significantly eroded provided that they have the percentage >60%. Similar concerns to “the most significant erosion”. 

Response 13: The results indicating erosion is based on three different methods where NSM and SCE are not reporting the rate value, but these methods represent the shoreline changes in distance value, while EPR result illustrated the trend of erosion and accretion that occur in the coastal zone. See line 186-187, 191-192. Sg. Nibong has significant accretion rate as the percentage >60%, highlighted in Table 4. Bagan Sg Burong and Sg Nibong has been analysed as the most significant accretion area, with the same percent of accretion, sentence has been revised. Please see line 373-375.

Reviewer 2 Report

This manuscript analysed the shoreline changes in 11 beaches along the Selangor coast (Malaysia), using DSAS (Digital Shoreline Analysis System, ESRI) software. The authors also presented a list of coastal adaptation measures implemented in the Selangor region.

The study is interesting since it analyses the historical and future behaviour of a wide coastal zone, however, I feel that the manuscript presents some lacks. Overall, the “Results and Discussion” section should be carefully revised, and a better-organized analysis of the behaviour of each site should be provided. I would suggest revising the paper as suggested in the following.

Main comments:

L78 In the “Study Area” section, I suggest providing a more quantitative description of the main factors that could affect the shoreline behaviour such as wave climate and tide, beach slope and sediments, vegetation, anthropogenic activities, presence of any nourishment and coastal protection structures. This information is relevant for the analysis of the results and would help the reader to understand the differences between the different study areas.

L171 The “Results and Discussion” section is not well organized and difficult to read, and the final part (L298-335) seems disconnected from the previous results. I suggest rewriting the chapter by paying attention to i) analysing the behaviour of each study area considering the factors that may have affected or could affect the evolution of the coast in the future, and ii) comparing the results obtained with the three different methods. Moreover, to make the results more reliable, it might be useful to evaluate the presence of cyclical or seasonal behaviours and, if relevant, the effect of the tide on shoreline detection.

Line specific comments:

L84 Change “government” to “Government”.

L113 Change “computation” to “computation.”.

L114 Adding an image with the shorelines extracted could help the reader to better understand the dataset.

L116 The baseline was drawn parallel to which shoreline?  Was the shoreline analysis performed separately for each study site?

L117 What do you mean by “transect line”? If you mean the generated transects, change to “transect lines”. In general, along the manuscript, "transect lines" could be changed to “transects”.

L126 Is fx the distance of the further shoreline from the baseline? If yes, add “further”.

L143 What shorelines have been considered for A and B? Are they the youngest and oldest shorelines?

L150 Add reference for Fenster et al, 1993.

L 164 Check the equation. Is MEPR equal to EPR? Is it (S1-S2) or (S2-S1)?

L166 Change “T” to “T1”.

L167-168 This sentence is unclear, please reword it. Is P (maybe equal to p) the predicted shoreline position?

L172-175 I would like to delete this part or move it to the introduction section.

L178-179 Are the results reported in Table 3 the min and max distance considering all the transects inherent to the same site?

L182-185 Please specify these statements better. It is not clear to the reader which study areas these comments refer to.

L193 Bagan Sg. Burong has the maximum accretion.

In tables 4 and 5, please specify in the “Shoreline analysis” section how the reported values have been computed.

L251 Have two rates of shoreline changes been computed using the EPR method? If yes, which was used for the prediction analysis and why?

L275-281 This part sound more like a conclusion, consider moving it to the “Conclusion” section.

L282-289 Move this part to the “Study Area” section.

L290-297 Consider moving this part to the “Introduction” section.

L309 Add reference for Cao et al..

L298-335 Clarify why this part is linked to the paper results. Consider summarizing it in the “Introduction” section.

Author Response

Reviewer 2

This manuscript analysed the shoreline changes in 11 beaches along the Selangor coast (Malaysia), using DSAS (Digital Shoreline Analysis System, ESRI) software. The authors also presented a list of coastal adaptation measures implemented in the Selangor region.

The study is interesting since it analyses the historical and future behaviour of a wide coastal zone, however, I feel that the manuscript presents some lacks. Overall, the “Results and Discussion” section should be carefully revised, and a better-organized analysis of the behaviour of each site should be provided. I would suggest revising the paper as suggested in the following.

Response: Thank you for the developing comments. We have revised the paper according to the comments below.

Main comments

Point 1: L78 In the “Study Area” section, I suggest providing a more quantitative description of the main factors that could affect the shoreline behaviour such as wave climate and tide, beach slope and sediments, vegetation, anthropogenic activities, presence of any nourishment and coastal protection structures. This information is relevant for the analysis of the results and would help the reader to understand the differences between the different study areas.

Response 1: General info describing the study area has been added as suggested. Please see line117-129.

Point 2: L171 The “Results and Discussion” section is not well organized and difficult to read, and the final part (L298-335) seems disconnected from the previous results. I suggest rewriting the chapter by paying attention to i) analysing the behaviour of each study area considering the factors that may have affected or could affect the evolution of the coast in the future, and ii) comparing the results obtained with the three different methods. Moreover, to make the results more reliable, it might be useful to evaluate the presence of cyclical or seasonal behaviours and, if relevant, the effect of the tide on shoreline detection.

Response 2: The results and discussion has been rewritten accordingly. The three different methods have been compared in last part of discussion as stated in line 347-366. The presence of seasonal behavior is not relevant in this study as this study focused on data from January to June in which the area has the same seasonal behavior. Recommendation has been stated in the conclusion part for future study considering the seasonal behaviour. Line 378-380.

Specific comments

Point 1: L84 Change “government” to “Government”.

Response 1: The term has been corrected. Please see line 115.

Point 2: L113 Change “computation” to “computation.”.

Response 2: Changed, see line 159.

Point 3: L114 Adding an image with the shorelines extracted could help the reader to better understand the dataset.

Response 3: Figures on the shoreline extracted from 1993 to 2004 and 2004 until 2014 has been added. Please see Figure 6 and Figure 8.

Point 4: L116 The baseline was drawn parallel to which shoreline?  Was the shoreline analysis performed separately for each study site?

Response 4:  The baseline was drawn parallel to shoreline in 1993. The analysis was performed at the same time for each study site. See line 163-164.

Point 5: L117 What do you mean by “transect line”? If you mean the generated transects, change to “transect lines”. In general, along the manuscript, "transect lines" could be changed to “transects”.

Response 5: The term has been changed to transects accordingly. See line 161,166,180,185,225,236.  

Point 6: L126 Is fx the distance of the further shoreline from the baseline? If yes, add “further”.

Response 6: The sentence has been revised as in line 176.

Point 7: L143 What shorelines have been considered for A and B? Are they the youngest and oldest shorelines?

Response 7: The sentence has been edited accordingly. See line 195-196.

Point 8: L150 Add reference for Fenster et al, 1993.

Response 8: Numbering citation has been added. Please see line 202.

Point 9: L 164 Check the equation. Is MEPR equal to EPR? Is it (S1-S2) or (S2-S1)?

Response 9: MEPR refers to the rate of shoreline, is not equal to EPR. Revised to (S2-S1), please see line 215.

Point 10: L166 Change “T” to “T1”.

Response 10: Changed, see line 217.

Point 11: L167-168 This sentence is unclear, please reword it. Is P (maybe equal to p) the predicted shoreline position?

Response 11: Yes, its predicted shoreline position, the term has been revised. See line 218.

Point 12: L172-175 I would like to delete this part or move it to the introduction section.

Response 12: The sentence has been deleted as it has been stated in the introduction. Please see line 58-60.

Point 13: L178-179 Are the results reported in Table 3 the min and max distance considering all the transects inherent to the same site?

Response 13: Table 3 is based on SCE which measures the total change in coastline considering all available coastline positions and reports on their distance, inherent to the same site without reference to a specific date,

Point 14: L182-185 Please specify these statements better. It is not clear to the reader which study areas these comments refer to.

Response 14: The statement has been edited to refer to the area referred to. See line 229-232.

Point 15: L193 Bagan Sg. Burong has the maximum accretion.

Response 15: Sentence has been revised as suggested. See line 243.

Point 16: In tables 4 and 5, please specify in the “Shoreline analysis” section how the reported values have been computed.

Response 16: Values in Table 4 was computed from Equation 2 while values in Table 5 was derived from Equation 3. See line 240, 284-286.

Point 17: L251 Have two rates of shoreline changes been computed using the EPR method? If yes, which was used for the prediction analysis and why?

Response 17: Yes, the prediction for year 2030 and 2040 has been computed using EPR method. The validation of EPR model was done by comparing the positional difference from the extracted shoreline of 2014. Models based on EPR has been widely used in the predication of future temporal changes of the shoreline. Those models assume that the historical observed change rates of the shoreline can capture the cumulative effects of coastal processes such as waves, currents and storms. The basic assumption of EPR model is that the observed historical rates of the shoreline changes are the best estimates for predicting its future position. Please see line 309-315.

Point 18: L275-281 This part sound more like a conclusion, consider moving it to the “Conclusion” section.

Response 18: The sentence has been moved to conclusion section. Please see line 369-371.

Point 19: L282-289 Move this part to the “Study Area” section.

Response 19: The mentioned part has been moved to study area, see line 117-124.

Point 20: L290-297 Consider moving this part to the “Introduction” section.

Response 20: The sentence has been removed to Introduction section. Please see line 65-80.

Point 21: L309 Add reference for Cao et al..

Response 21: Numbering citation has been updated in the sentence. Please see line 72.

Point 22: L298-335 Clarify why this part is linked to the paper results. Consider summarizing it in the “Introduction” section.

Response 22: The sentence has been reconstructed to Introduction part as it is not part of the results. Please see line 68-80.

Reviewer 3 Report

While the current paper is trying to study changes in the Selangor coast's shoreline, the lack of a solid literature review to support the outputs from DSAS can be seen. The authors have tried to address this by providing some proofs from different sources but without establishing a well-organized literature review, it is hard to ensure the reliability of the DSAS outputs. It should not be neglected that the DSAS is just a statistical approach with many limitations (as the authors have mentioned some of them on page 5, line 154). Moreover, the uncertainties related to the DSAS (e.g., Digitized uncertainty, Proxy-Datum uncertainty, etc.) are missed. The followings are some other comments on each section:

- Introduction: 

Motivations for conducting the research could be better explained in the introduction. While there are a couple of phrases indicating the issue on a global scale, a lack of information about the studied area can be seen. Moreover, reviewing prior research related to the studied area (if there is any) could enrich the introduction section. 

Line 53: It seems that the word "growth" is missed after "population"

The following sentences need to be rewritten more concisely to better convey their messages: 

Line 67: "Nowadays, upgrading technologies are more useful and valuable for conducting 67 coastal studies".

Line 71: "DSAS is a tool to compute the changes using a statistical approach". For example "... to compute rate-of-changes ..." or "... to study shoreline changes ..."

- Study Area: More specific and unique information related to the studied area should be given. Providing the coordinates of the sites is not enough. In addition, Line 83-84 could be stated in the introduction.

- Data sources:

Table 2: Why did you name the second column "Parameters"? 

Table 2: If you insist to merge the spatial resolution and the scale in one column, at least you should note it in the text or present it in a better way so the reader can understand it without too much effort.

- Shoreline analysis:

Efforts of the authors for providing information about SCE, NSM, and EPR are appreciated, but since the DSAS is not anymore an unknown tool for the researchers and lots of research are published using this technique, it is highly recommended to considerably summarize this information in order to save words. Instead, the authors can develop the introduction and the Study Area section.  

Line 114: More information about the technique used for digitizing should be given (It is very important!). For example, which proxy was used for extracting the vector shoreline position?

- Shoreline changes prediction: 

Lines 150-152: How do you evaluate the EPR compared to WLR? What are the reasons for not employing the WLR?  

- Results and Discussion:

Lines 172-177: They seem to be redundant here. They can be either omitted or moved to the Conclusion section.  

Lines 195-197: If there are some studies that can be used for verifying the results (e.g., Selamat et al. & Asmawi & Ibrahim), they should be mentioned in the text as the literature review.   

Page 11: Again, this information should be given in the introduction unless the authors can directly relate them with the outputs of the DSAS.

- Conclusion:

After making the recommended changes, this section should also be revised accordingly.

Author Response

Reviewer 3

Point 1: While the current paper is trying to study changes in the Selangor coast's shoreline, the lack of a solid literature review to support the outputs from DSAS can be seen. The authors have tried to address this by providing some proofs from different sources but without establishing a well-organized literature review, it is hard to ensure the reliability of the DSAS outputs. It should not be neglected that the DSAS is just a statistical approach with many limitations (as the authors have mentioned some of them on page 5, line 154). Moreover, the uncertainties related to the DSAS (e.g., Digitized uncertainty, Proxy-Datum uncertainty, etc.) are missed. The followings are some other comments on each section:

-Introduction: 
Motivations for conducting the research could be better explained in the introduction. While there are a couple of phrases indicating the issue on a global scale, a lack of information about the studied area can be seen. Moreover, reviewing prior research related to the studied area (if there is any) could enrich the introduction section. 

Response 1: Thank you for the comments. The mentioned comments are revised as stated below. Introduction has been edited with information on study area accordingly. Please see line 58-96.

Point 2: Line 53: It seems that the word "growth" is missed after "population"

Response 2: Word growth has been added accordingly. See line 53.

Point 3: The following sentences need to be rewritten more concisely to better convey their messages: Line 67: "Nowadays, upgrading technologies are more useful and valuable for conducting 67 coastal studies". Point 4: Line 71: "DSAS is a tool to compute the changes using a statistical approach". For example "... to compute rate-of-changes ..." or "... to study shoreline changes ..."

Response 3: Both sentences has been revised as suggested. Please see line 97,101-102.

Point 4: Study Area: More specific and unique information related to the studied area should be given. Providing the coordinates of the sites is not enough. In addition, Line 83-84 could be stated in the introduction.

Response 4: Information on study area has been updated and edited, line 117-129. The suggested sentence has been moved to introduction, see line 59-60.

Point 5: Data sources:Table 2: Why did you name the second column "Parameters"?  Table 2: If you insist to merge the spatial resolution and the scale in one column, at least you should note it in the text or present it in a better way so the reader can understand it without too much effort.

Response 5: The parameter label in Table 2 has been corrected. Please see line 155.

Point 6: Shoreline analysis: Efforts of the authors for providing information about SCE, NSM, and EPR are appreciated, but since the DSAS is not anymore an unknown tool for the researchers and lots of research are published using this technique, it is highly recommended to considerably summarize this information in order to save words. Instead, the authors can develop the introduction and the Study Area section.  

Response 6: The sentence has been moved to Introduction. See line 58-65.

Point 7: Line 114: More information about the technique used for digitizing should be given (It is very important!). For example, which proxy was used for extracting the vector shoreline position?

Response 7: General information related to the study area has been added in line with previous reviewers’ comment. Please see line 117-129, 163-164.

Point 8: Shoreline changes prediction: Lines 150-152: How do you evaluate the EPR compared to WLR? What are the reasons for not employing the WLR?  

Response 8: Both EPR and WLR are acceptable for shoreline change analysis. Specifically, if uncertainties are known, WLR provide the best results. If uncertainties are unknown, the best method would be EPR as it requires only two shoreline dates for its computation, its inconvenience appears when there are many shorelines, all intermediate shorelines dates are ignored. Please see line 362-366.

Point 9 :Results and Discussion: Lines 172-177: They seem to be redundant here. They can be either omitted or moved to the Conclusion section.  

Response 9: The sentence has been moved to Introduction. Please see line 93-96.

Point 10: Lines 195-197: If there are some studies that can be used for verifying the results (e.g., Selamat et al. & Asmawi & Ibrahim), they should be mentioned in the text as the literature review.   

Response 10: These citations were included as to compare the results between the current study and past study. We believed these sentences should remain in the paragraph. Line 246-248.

Point 11: Page 11: Again, this information should be given in the introduction unless the authors can directly relate them with the outputs of the DSAS.

Response 11: The paragraph has been moved to introduction accordingly. See line 58-76.

Point 12: Conclusion: After making the recommended changes, this section should also be revised accordingly.

Response 12: Conclusion part has been edited as according to other reviewers. Please see line 369-385.

Round 2

Reviewer 3 Report

Dear authors,

The efforts you have made to address most of the previous comments are satisfactory and appreciated. Although, please pay attention to the following points:

Point 7: Line 114: More information about the technique used for digitizing should be given (It is very important!). For example, which proxy was used for extracting the vector shoreline position?

Response 7: General information related to the study area has been added in line with previous reviewers’ comment. Please see line 117-129, 163-164.

(New Comment): To better explain my point with this comment, please see the following statement from DSAS manual:

"Shoreline positions can reference several different features such as the vegetation linethe high water linethe low water line, or the wet/dry line. They can be digitized from a variety of sources (for example, satellite imagery, digital orthophotos, historical coastal-survey maps), collected by global positioning-system field surveys, or extracted from lidar surveys" (Himmelstoss, 2009, p.10).  

Himmelstoss, E.A. 2009. “DSAS 4.0 Installation Instructions and User Guide” in: Thieler, E.R., Himmelstoss, E.A., Zichichi, J.L., and Ergul, Ayhan. 2009 Digital Shoreline Analysis System (DSAS) version 4.0 — An ArcGIS extension for calculating shoreline change: U.S. Geological Survey Open File Report 2008-1278. *updated for version 4.3.

Considering this, which feature did you use as a reference? Please include it in the text.  

----------------------------

Point 8: Shoreline changes prediction: Lines 150-152: How do you evaluate the EPR compared to WLR? What are the reasons for not employing the WLR?

Response 8: Both EPR and WLR are acceptable for shoreline change analysis. Specifically, if uncertainties are known, WLR provide the best results. If uncertainties are unknown, the best method would be EPR as it requires only two shoreline dates for its computation, its inconvenience appears when there are many shorelines, all intermediate shorelines dates are ignored. Please see line 362-366.

(New Comment): For your own records, please notice that there are several publications in which the uncertainties are estimated. For example:

Hapke, Cheryl J., Emily A. Himmelstoss, Meredith G. Kratzmann, and E. Robert Thieler. "National assessment of shoreline change: Historical shoreline change along the New England and Mid-Atlantic coasts." (2011).

Rezaee, Seyed Meysam, Aliasghar Golshani, and Sahand Abedi. "Shoreline changes at Fereydounkenar Port in light of Caspian Sea’s water level fluctuations." Regional Studies in Marine Science 53 (2022): 102393.

Therefore, as a suggestion (optional), there is a possibility to improve your work in case you are interested to do so. In case you are not, the explanation you have provided in lines 362-366 seems to be enough. Although you should reconsider "the most suitable method" (mandatory!).

----------------------------

Point 10: Lines 195-197: If there are some studies that can be used for verifying the results (e.g., Selamat et al. & Asmawi & Ibrahim), they should be mentioned in the text as the literature review.

Response 10: These citations were included as to compare the results between the current study and past study. We believed these sentences should remain in the paragraph. Line 246-248.

(New Comment): It is true that they should remain in the paragraph that you are mentioning. But, my point is that providing previous works related to your area of study (in the introduction) would greatly enhance your paper (optional). You can indicate these citations there.

My best wishes!

Author Response

Dear reviewers

Warm regards,

I would like to thank the reviewer for considering our manuscript and for the valuable time exerted in reviewing it. We appreciate the constructive criticisms of this manuscript. We have addressed all issues indicated in the reviewer’s report point by point and highlighted the changes with the yellow color.

Dear authors,

The efforts you have made to address most of the previous comments are satisfactory and appreciated. Although, please pay attention to the following points:

Point 7: Line 114: More information about the technique used for digitizing should be given (It is very important!). For example, which proxy was used for extracting the vector shoreline position?

Response 7: General information related to the study area has been added in line with previous reviewers’ comment. Please see line 117-129, 163-164.

Point 1: (New Comment): To better explain my point with this comment, please see the following statement from DSAS manual:

"Shoreline positions can reference several different features such as the vegetation linethe high water linethe low water line, or the wet/dry line. They can be digitized from a variety of sources (for example, satellite imagery, digital orthophotos, historical coastal-survey maps), collected by global positioning-system field surveys, or extracted from lidar surveys" (Himmelstoss, 2009, p.10).  

Himmelstoss, E.A. 2009. “DSAS 4.0 Installation Instructions and User Guide” in: Thieler, E.R., Himmelstoss, E.A., Zichichi, J.L., and Ergul, Ayhan. 2009 Digital Shoreline Analysis System (DSAS) version 4.0 — An ArcGIS extension for calculating shoreline change: U.S. Geological Survey Open File Report 2008-1278. *updated for version 4.3.

Considering this, which feature did you use as a reference? Please include it in the text.  

Response 1: The vegetation line feature has been added in the paragraph and cited. Please refer line 164-165, 428-430.

----------------------------

Point 8: Shoreline changes prediction: Lines 150-152: How do you evaluate the EPR compared to WLR? What are the reasons for not employing the WLR?

Response 8: Both EPR and WLR are acceptable for shoreline change analysis. Specifically, if uncertainties are known, WLR provide the best results. If uncertainties are unknown, the best method would be EPR as it requires only two shoreline dates for its computation, its inconvenience appears when there are many shorelines, all intermediate shorelines dates are ignored. Please see line 362-366.

Point 2: (New Comment): For your own records, please notice that there are several publications in which the uncertainties are estimated. For example:

Hapke, Cheryl J., Emily A. Himmelstoss, Meredith G. Kratzmann, and E. Robert Thieler. "National assessment of shoreline change: Historical shoreline change along the New England and Mid-Atlantic coasts." (2011).

Rezaee, Seyed Meysam, Aliasghar Golshani, and Sahand Abedi. "Shoreline changes at Fereydounkenar Port in light of Caspian Sea’s water level fluctuations." Regional Studies in Marine Science 53 (2022): 102393.

Therefore, as a suggestion (optional), there is a possibility to improve your work in case you are interested to do so. In case you are not, the explanation you have provided in lines 362-366 seems to be enough. Although you should reconsider "the most suitable method" (mandatory!).

Response 2: The sentence has been changed to the most applicable method based on this study. Line 367-368.

----------------------------

Point 10: Lines 195-197: If there are some studies that can be used for verifying the results (e.g., Selamat et al. & Asmawi & Ibrahim), they should be mentioned in the text as the literature review.

Response 10: These citations were included as to compare the results between the current study and past study. We believed these sentences should remain in the paragraph. Line 246-248.

(New Comment): It is true that they should remain in the paragraph that you are mentioning. But, my point is that providing previous works related to your area of study (in the introduction) would greatly enhance your paper (optional). You can indicate these citations there.

Response 3: Related previous work has been added in the paragraph. Please see line 250-253.
